# An Integrated Model for Evaluating the Sustainability of Gamified Mobile Health Apps: An Instrument Development and Validation

**DOI:** 10.3390/healthcare11071051

**Published:** 2023-04-06

**Authors:** Abdulsalam Salihu Mustafa, Nor’ashikin Ali, Jaspaljeet Singh Dhillon, Darshana Sedera

**Affiliations:** 1College of Graduate Studies, Universiti Tenaga Nasional, Kajang 43000, Malaysia; 2College of Computing and Informatics, Universiti Tenaga Nasional, Kajang 43000, Malaysia; 3Faculty of Business, Law and Arts, Southern Cross University, East Lismore, NSW 2480, Australia

**Keywords:** gamification, mHealth, physical activity, motivation, continuance intention

## Abstract

Mobile health (mHealth) apps are designed to support health behavior outcomes and improve well-being. The existing body of literature confirms mHealth’s overall efficacy in promoting physical activity; however, more research on its utility in sustaining user engagement is needed. Understanding the determinants of an individual’s willingness to continue using mHealth is vital to improving the intervention’s success. This study developed a unified model and survey instrument adapted from extant literature while introducing new constructs to predict the sustained use of gamified mHealth. A pilot study was conducted to validate the survey instrument using 48 gamified fitness app users in Malaysia. The survey instrument was tested following rigorous guidelines for quantitative research in the information system context. According to the findings, the reliabilities of most measurement items met the criterion, and those items were retained. Overall, this paper contributes by integrating social comparison theory and the self-determination theory for sustaining user engagement with gamified mHealth through an extrinsic and intrinsic motivation perspective.

## 1. Introduction

In health and fitness, sustained physical activity can enhance our quality of life and well-being. The advent of technology in our daily activities has decreased people’s engagement with physical activity [1]. Thus, this critical component of a healthy lifestyle has diminished, increasing the risk of chronic diseases such as obesity, which can negatively affect health and well-being, not to mention the increase in public healthcare costs. In Malaysia, national statistics report that about 50.1% of adults are overweight or obese [2]. This is attributed to several risk factors, such as unhealthy lifestyle, environment, age, family history, genetics, race, and ethnicity [3,4]. Predominantly, improper nutrition and physical inactivity are among these unhealthy lifestyle habits. Physical inactivity, such as sedentarism, increases the risk of being overweight. A sedentary lifestyle involves sitting or lying down for lengthy periods [5]. Hence, sedentary habits remain a significant public health risk worldwide.

Studies suggest a strong correlation between physical activity and well-being [6]. Simply put, inadequate physical activity can be detrimental to our health. The World Health Organization (WHO) recommends at least 150 or 75 min of moderate or vigorous workouts weekly to remain physically active. Even so, increasing evidence shows that many people do not adhere to the guidelines, with a 25% compliance rate worldwide [4]. The 2019 data [2] revealed that 21.5% of Malaysians were physically inactive. Thus, promoting compliance with physical activity guidelines is critical to good health and well-being.

To this end, health concerns have driven the adoption of various health interventions for disease prevention and treatment. Health interventions, popularly known as mobile health (mHealth) apps, offer cost-effective, cutting-edge approaches to medication and intervention design [7]. According to the WHO [8], mHealth refers to using mobile devices to support and deliver public healthcare. mHealth can be considered a subcategory of eHealth, a broader term for using electronic technologies such as computers in public health. Such interventions primarily target dietary and physical activity enhancement through behavior change strategies such as goal setting, self-monitoring, and prompting [9]. Smart mobile devices have rapidly become the primary key enabler for health interventions, resulting in increased adoption of health apps. They deliver real-time data, improve healthcare access, and increase dissemination capability [10]. Additionally, mHealth facilitates timelier access to health-related information, medical advice, or services regardless of location [11].

There are several health apps available in the app stores. A vast majority of those apps are free to download and use but offer basic features. Other apps require a monthly or annual subscription to access advanced features. The most popular mHealth apps track physical activities, diet, and medication adherence [12]. Those apps include Gymshark Training, Jetfit, MDCalc, mSafety, Ovia Health, and Unmind. In 2017, over 325,000 health apps were available in major app stores, representing an impressive increase of 78,000 since 2016 [13]. Additionally, mHealth was valued at about USD 28 billion in 2018 and USD 56 billion in 2020 and is expected to reach USD 151.6 (26% increase) in 2025 [14].

The numerous benefits of mHealth give rise to a growing interest in its adoption. A recent empirical study on mHealth usage patterns in Malaysia reported a high acceptance rate among smartphone users [15]. However, its benefits are coupled with unique challenges. A growing concern in this domain is the high attrition rates, as users tend to quit the apps before reaching their fitness goals [16]. Individuals end up utilizing the apps less after the initial or post-adoption stage due to losing interest and excitement, resulting in app abandonment. Thus, continuance intention is an important driving force for the long-term use of mHealth.

To address this issue critically, developers employed gamification as a tool that attempts to sustain user engagement by transforming boring activities related to exercising or dieting into fun experiences [17]. Games are generally designed to keep players engaged and in a state of flow; gamification builds on this concept by leveraging game elements and mechanics in a non-game domain [18]. In this specific context, the non-game domain relates to using mHealth to encourage regular exercise. This is usually referred to as the gamification of mHealth or gamified mHealth. The game-design elements in those health apps provide intrinsic or extrinsic motivational affordances [19].

Not surprisingly, there has been a surge in gameful designs in healthcare to encourage maintaining a healthy lifestyle. Nonetheless, rapid development in this field has resulted in a widening gap between consumer apps and scientific bases. For example, empirical studies show that some health apps do not adequately meet users’ requirements [16]. Consequently, those gamified health apps failed to motivate users to continue using them in the post-adoption stage [20,21]. Therefore, it is imperative to identify factors predicting sustained engagement with gamified mHealth.

It is generally observed that the discontinued use of mHealth can negatively affect users’ health-related behaviors [22]. The body of research constantly shows that user abandonment of mHealth remains a significant concern for app developers, with about 27.8% of mHealth apps uninstalled within 30 days of their download [23]. In particular, a lack of motivation can prompt users to lose interest in engaging with mHealth. Ultimately, they may withdraw from the health goals they are trying to achieve via mHealth, leading to notable high drop-out rates. This is a critical question, as research on using gamified mHealth beyond the adoption phase is sparse; most studies use well-grounded theories with several limitations [24,25,26]. For example, single theories fail to effectively explain user motivation and the psychological aspects of health behavior change [24,25]. Most notably, those theories hardly distinguish the varied effect of intrinsic and extrinsic motivation on continued use behavior. Across the body of research, it is glaring that more attention needs to be given to the impact of individual game elements on the sustained use of mHealth, as most studies examined their combined effect [26]. Thus, there is limited evidence of individual game elements’ intrinsic or extrinsic effects on sustained user engagement with mHealth.

Currently, there is inconclusive evidence surrounding understanding how game-like features in mHealth effectively sustain healthy behaviors [26,27]. Most studies approach gamification as a single concept using similar game elements without a defined standard in the extant literature. However, game elements are not generic and can produce varied effects depending on their combinations and context [26]. Thus, it is challenging to determine which elements to combine to sustain user engagement. Instead, identifying predictors of successful outcomes in gamified mHealth is crucial to improving its effectiveness, engagement, and guiding approaches to sustaining long-term behavior change. In this sense, the first step involves developing a theoretical model and validating the survey instrument.

This study is structured as follows. Section 2 covers the research model and proposed hypotheses of the study. Section 3 presents the survey instrument. Section 4 explains the research methodology. Section 5 includes internal validity and consistency results. The results are discussed in Section 6. Finally, Section 7 points out the theoretical implications of the study findings and future research directions.

## 2. Research Model and Hypotheses Development

Different theoretical lenses have been applied to explain and predict the continuance intention toward mHealth, including the Technology Acceptance Model (TAM), Social Cognitive Theory (SCT), Self-Efficacy Theory (SET), Uses and Gratifications (U and G), and Task-Technology Fit (TTF). Thus far, prior studies adopting the above-stated theoretical lenses mainly focused on mHealth adoption, with limited attention given to the continued use of gamified mHealth. As noted earlier, existing theories do not sufficiently explain the different effects of game elements in mHealth [24]. In addressing these gaps, one potential research avenue is extending theories with external variables or integrating them with other theories [28,29]. Theoretical integration combines the strengths of different theories to address their limitations [28]. This concept, popularized by Hagger [30] in healthcare, has gained prominence as a tool for better understanding complex health-related behaviors. Theoretical integration is crucial as they underline the essential psychological variables that predict and explain health behavior while eliminating redundancy [30]. For instance, well-established theories in the literature, such as TAM or TTF, have been integrated to address their underlining weaknesses. 

With that in mind, this study adopted a multi-theory, integrated approach by implementing a framework integrating theories from psychology, behavioral science, and game-like features to improve users’ adherence to mHealth. The proposed model is grounded in the integration of the self-determination theory (SDT) [31] and social comparison theory (ScT) [32], focusing on intrinsic and extrinsic motivational styles. In summary, SDT provides insights on intrinsic motivation, ScT on extrinsic motivation, external variables on utilitarian, hedonistic, and environmental perspectives, and game features on the different motivational affordances. This study aims to thoroughly understand factors influencing the motivation to sustain using gamified mHealth for healthy habits, which is essential to preventing or managing obesity. Using extant literature, a research model was developed.

In the context of this study, the psychological and behavioral outcomes are centered on the continued use of mHealth. Studies affirm that healthy behaviors require psychological and behavioral viewpoints [33]. The psychological aspect relates to autonomy, competence, relatedness, hedonic gratification, and social comparison. Additionally, new constructs, fitness app quality, perceived benefits, and facilitating conditions are included. Fitness app quality and perceived benefits have been proven to satisfy utilitarian motives for using fitness apps [34]. Facilitating conditions positively influence behavioral intention in the health context [35].

### 2.1. Self-Determination Theory

SDT is a well-established motivational theory for understanding an individual’s intrinsic and extrinsic motives for engaging in behavior [28]. Essentially, the theory emphasizes the human desire to satisfy three core psychological needs: autonomy, competence, and relatedness [31]. Autonomy refers to having choices, competence relates to mastery, and relatedness pertains to social connectedness. SDT models a correlation between individuals’ basic desires, social factors, and conscious processes [36]. The theory distinguishes three types of motivation along a continuum focusing on the psychological level: amotivation, and extrinsic and intrinsic motivation. Individuals can progress from a non-autonomous or controlled form of motivation toward a more autonomous form of motivation [37]. Thus, people can initiate an activity for external reasons; over time, they may derive pleasure from it and internalize it, thus becoming intrinsically motivated. Researchers conclude that SDT is a practical framework for collecting and examining motivation and behavioral change data [24,27]. The theory is widely applied in the gamification literature to provide insights into the motivational effects of game elements, primarily intrinsic. SDT’s suitability for gamification is because it helps us understand how people are motivated in a continuum [38]. Thus, the process by which a person moves through the different motivational stages (Controlled to Autonomous motivation). In addition, it addresses the psychological effects of game elements. Those effects determine if a person will be motivated or not. However, SDT has failed to clearly distinguished the impact of individual and combined game-like elements on sustaining user motivation and health-related behavior [39]. Hence, it necessitates combining SDT with similar theories and their relevant features into a consistent basis for a deeper understanding.

### 2.2. Social Comparison Theory

ScT is based on the notion that people can reliably evaluate their thoughts and capabilities [32]. The theory applies social comparison and self-enhancement to increase motivation [32]. For instance, ScT can complement gamification features by fostering competition through leaderboards and encouraging challenges for self-improvement. Studies implementing game-like features to facilitate social comparison recognized its significant effect on extrinsic motivation. In this sense, Huschens and co-authors [40] found leaderboards essential in effectively encouraging social comparison and improving staff motivation. They concluded that the game elements directly shape social comparison behavior, an important channel by which gamification unfolds motivational and pressure effects on users. Similarly, Landers et al. [24] reported that leaderboards significantly increase a task’s short-term performance. Even though those studies provide evidence of ScT’s ability to motivate individuals extrinsically, they could not sustain long-term user engagement. According to logic, extrinsic motivation effectively encourages individuals to initiate a behavior or activity; however, it may fail to sustain it [41]. Thus, it was inferred that the positive effect of these investigations was short-term, and user motivation decreased over time. Accordingly, combining ScT with other theories that support intrinsic motivation is crucial in empirically determining its relationship with various game elements in the gamification context.

Based on the above, the proposed integrated theory would help to compensate for the limitations of the SDT theoretical framework and enrich our understanding of the antecedents of mHealth’s continued use. Figure 1 illustrates the proposed model, which combines SDT concepts of perceived autonomy, competence, and relatedness with ScT. External variables include fitness app quality, perceived hedonic gratification, perceived benefits, and facilitating conditions. Moreover, gamification variables include feedback, leaderboards, challenge, progress bars, and levels. Therefore, developing items to assess the constructs to test the proposed model empirically is critical. 

### 2.3. Perceived Autonomy (PA)

Perceived autonomy (PA) is essential in explaining individuals’ continuance intention. According to SDT, autonomy refers to the need to self-regulate an individual’s behavior and experience [42]. More specifically, autonomy describes the extent to which people perceive that they can determine their actions rather than being pressured or regulated [31]. Significantly, PA depends on the level of control players have and the various options available during gameplay [43]. Autonomy is fundamental to the intrinsic and self-determined forms of motivation. Previous healthcare studies found that PA is critical to user intention and motivation to exercise [39,44]. That is, when individuals realize a greater sense of autonomy and control, they are more motivated to engage in a behavior. On the other hand, intrinsic motivation decreases when users feel controlled or less autonomous [31]. Thus, gamification leverages this by offering a degree of autonomy to users, giving them the perception that they have control over their actions. From our study’s perspective, we believe the need for autonomy will positively affect users’ motivation to continue using a gamified health app. It leads to the following hypothesis:

**H1.** 
*Perceived Autonomy has a positive influence on intrinsic motivation to continue using gamified mHealth apps.*


### 2.4. Perceived Competence (PC)

Competence from SDT’s perspective refers to the need to achieve desired results and avoid unwanted outcomes [37]. Perceived competence (PC) is associated with mastery and a perception of a person’s effectiveness in a task or activity [39]. Equally important, it entails a person’s need to feel effective at daily activities and gain competence over time. Prior studies provide empirical evidence establishing a positive association between PC, intrinsic motivation, and adherence [45,46]. It implies that an increased feeling of competence can lead to self-determined motivation and, eventually, sustained behavior change. In the physical activity context, individuals who do not feel confident exercising may have lower PC and ultimately become less motivated or amotivated to be physically active. Subsequently, mHealth users experiencing low motivation may perceive the app as a barrier and eventually stop using them. Contrastingly, increased motivation can inspire users to continue exercising [47]. In line with SDT, we can infer that gamified mHealth will enable users to experience PC and motivate them to continue engaging. Thus, we posit that:

**H2.** 
*Perceived Competence has a positive influence on intrinsic motivation to continue using gamified mHealth apps.*


### 2.5. Perceived Relatedness (PR)

According to SDT, relatedness applies to a person’s desire to connect with others [42]. Perceived relatedness (PR) provokes individuals to feel socially integrated and part of a social community [31]. In sports, for example, players develop strong relationships with their teammates, reinforcing their desire to perform better. Along this line, researchers conclude that the desire to maintain players’ relatedness positively correlates with enjoyment and intrinsic motivation to continue playing [39]. Thus, when game players relate to and socialize, they are more driven to keep playing. Moreover, another study reported that students found PR intrinsically motivating when utilizing an e-learning program [48]. In fitness apps, essential game elements enable users to connect, support, and interact with others (family, friends, colleagues, and peers) on social media, supporting the need to satisfy PR. Tsai et al. [49] found social media usage and interaction to reinforce users’ intrinsic motivation to exercise and adhere to healthy behaviors. Given the preceding information, it is logical to assume that users’ basic need for PR when using gamified mHealth will positively affect their intrinsic motivation to sustain using it. Hence, the following hypothesis is put forward:

**H3.** 
*Perceived relatedness has a positive influence on intrinsic motivation to continue using gamified mHealth apps.*


### 2.6. Fitness App Quality (FAQ)

Fitness App Quality (FAQ) pertains to the quality of an app’s design in terms of its usability and features [12]. In this context, mHealth quality relates to a user’s positive assessment of the app’s features, ensuring it addresses user demands [50]. Users are more likely to use the app when comfortable with its design and technical features. Additionally, the FAQ includes the aspect of technical adequacy, content, and appearance. Prior studies on IS demonstrated the underlying effect of system quality on user behavior. For example, Palos-Sanchez et al. [51] noted that when users are satisfied with the quality of an IS, they perceive it as beneficial and are more inclined to continue utilizing it. However, this benefit has an extrinsic value. Then again, mHealth quality issues such as automation, response time, or interactivity may result in users abandoning the app [51]. In this line, if mHealth is high-quality, users will perceive it as beneficial. At the same time, FAQ can positively affect users’ extrinsic motivation to sustain using mHealth. Thus, we propose the following hypotheses:

**H4a.** 
*Fitness app quality will have a positive effect on extrinsic motivation to continue using gamified mHealth apps.*


**H4b.** 
*Fitness app quality will have a positive effect on the perceived benefit of using gamified mHealth apps.*


### 2.7. Perceived Benefits (PB)

Perceived benefits (PB) relate to the perception of the positive outcomes of a specific behavior [52]. Those positive outcomes could be health-related, such as treating cardiovascular or achieving physical well-being. In the healthcare context, PB explains an individual’s motives for behavior change, such as mHealth adoption or medication adherence. Previous studies reported that PB is also significant in individuals adopting and maintaining healthy behaviors [53]. For instance, when people perceive the barriers as greater than the health benefits, they may be discouraged from engaging in healthy behavior, notwithstanding their advantages. On another note, users would be inspired to engage in the behavior if the benefits are more significant than the barriers. Regarding the gamification of mHealth, PB reflects the subjective assessment of user engagement with a gamified health intervention to increase their motivation to exercise. Moreover, Wei et al. [54] revealed that PB significantly affects mHealth usage behavior. Given that those studies failed to determine the continued usage intention, in this present study, we presume that the probability that individuals will continue to engage with mHealth depends on their beliefs about its benefits. However, those benefits will motivate users extrinsically. Therefore, we put forward the following hypothesis:

**H5.** 
*Perceived Benefits will have a positive effect on extrinsic motivation to continue using gamified mHealth apps.*


### 2.8. Perceived Hedonic Gratification (PHG)

Hedonic gratification aims to stimulate humans psychologically to become self-fulfilled [55]. The hedonic factors strongly relate to fun, entertainment, and enjoyment [55]. When individuals intend to utilize a system, they expect it to be engaging, fun, and pleasant. Prior empirical studies found that people are more likely to continue using a system if they perceive it as fun, satisfying, or entertaining [56]. Effectively, hedonic-related motivation can intrinsically motivate individuals to continue using IS. SDT notes that intrinsically motivated individuals engage in activities they believe as inherently satisfying [57]. Consequently, when users enjoy using gamified mHealth, they become intrinsically motivated and sustain its usage to gain the intended benefits. Thus, the following hypothesis is put forward:

**H6.** 
*Perceived Hedonic Gratification has a positive effect on intrinsic motivation to continue using gamified mHealth app.*


Accordingly, it is critical to establish various game-design features and distinguish them so that future research can identify which combination of game features contributes to sustaining user motivation, enjoyment, and health-related benefits. To identify suitable gamification design elements, we carried out a prior study [58] that identified five predominant game components among gamified mHealth users: challenges, progress bars, leaderboards, levels, and feedback.

### 2.9. Challenge (CH)

Challenges motivate users to complete a task while engaging in it [59]. This game feature enables participants to test their abilities toward achieving mastery. However, overcoming challenges require time, expertise, effort, and creativity [60]. Those factors are crucial in determining a task’s difficulty level, which can affect user motivation regarding progress. To this end, players expect gamified systems to be more challenging over time toward achieving mastery; otherwise, they might get bored and abandon them [61]. In this sense, app designers leverage challenge game mechanics to produce intended behaviors and evoke emotions. In the gamified mHealth domain, challenges help users satisfy their intrinsic need for competence after completing a demanding task. The limited literature suggests that challenges positively relate to intrinsic need satisfaction in a gamified system [62]. Moreover, challenges have been shown to assist users in reaching their fitness goals in mHealth [63]. Therefore, in applying this concept, we expect gamified mHealth to challenge users to enhance their intrinsic motivation and continue engaging with it. Thus, the following is proposed:

**H7.** 
*Challenge has a positive effect on intrinsic motivation to continue using gamified mHealth apps.*


### 2.10. Levels (LV)

Levels show user progression or a player’s current position in a game at a certain period [60]. In game design, levels are not linear; their design, complexity, and difficulty may increase exponentially through each stage and decrease over time [64]. This means as players progress through the levels, the game’s difficulty increases with each new stage. It can inspire players to remain interested and engaged with the game to advance to the subsequent levels. Nevertheless, excluding levels from a gamified system can lead to users losing interest as they cannot achieve a measurable sense of progress [60]. Levels have been integrated into mHealth, where the user’s exercise routine depends on the intensity. For instance, some fitness apps utilize beginner, intermediate, and advanced user levels to determine the exercise routine [65]. Users can leverage this to assess their level of competence. Interestingly, lower levels may not require much effort or competence, whereas higher levels necessitate higher proficiency and action. However, it is an extrinsic incentive [66] and is more likely to influence extrinsic motivation in a gamified mHealth. Based on the above discussion, we propose:

**H8.** 
*Levels significantly affect extrinsic motivation to continue using gamified mHealth apps.*


### 2.11. Feedback (FB)

In gameplay, players’ achievement status is represented as medals, badges, or points when they complete challenges and activities [59]. In short, feedback informs users of their progress and subsequent action. Werbach and Hunter [60] note that feedback offers players a better gaming experience by indicating their performance. The game element informs users of their accomplishments and makes them feel more connected to the system. Notably, feedback is context-dependent and can be either positive or negative. Previous research theorized that negative feedback stimulates short-term achievement, whereas positive feedback fosters long-term performance through the need for autonomy and competence [66]. Feedback is generally perceived as controlling or informational [66]. However, feedback showing outcomes in mHealth is mainly controlling and extrinsic. As a result, it could thwart a user’s autonomy, diminishing intrinsic motivation. In this study, we apply a similar concept: feedback can affect extrinsic motivation by influencing users’ self-determined need for autonomy. This leads to the following hypothesis:

**H9.** 
*Feedback has a positive effect on extrinsic motivation to continue using gamified mHealth apps.*


### 2.12. Progress Bars (PGB)

Progress bars visually represent the advancement made toward goal attainment [61]. In gameplay, progress bars enable users to keep track of their points and proximity to the next milestone or level. It motivates them to stay focused on the next goal while progressing. The PGB uses feedback loops to regulate behavior in a particular direction [60]. A notable example is tracking daily blood pressure readings indicates a healthy or unhealthy lifestyle. Moreover, empirical results demonstrated the motivating effect of feedback from a progress bar to complete a task perceived as challenging, such as a survey or online resume [67]. Users enjoy reinforcement of their performance in an activity through a progress indicator. In gamification for mHealth, a progress indicator allows users to see how close they are to task completion. Inevitably, this may reinforce their healthy behaviors, but it could be short-term. Therefore, we expect the progress bar to positively influence the extrinsic motivation to continue using gamified mHealth and propose that: 

**H10.** 
*Progress Bars have a positive effect on extrinsic motivation to continue using gamified mHealth apps.*


### 2.13. Leaderboards (LB)

Leaderboards display users’ ranking relative to their peers in a table format [25]. It may function as a feedback system that fosters competitiveness to increase user engagement. Essentially, the LB is widely employed in gamified fitness apps to increase motivation and engagement through social comparisons and competitions [19,20,32,61]. However, the motivational effect is extrinsic [63]. Leaderboards are commonly associated with points and badges, referred to as PBL. Some researchers analyzed the impact of leaderboards on individuals and found leaderboards useful in facilitating user engagement [61]. For instance, Jia et al. [68] noted that students rated LB favorably in a fitness app. Similarly, Huschens et al. [40] studied the effect of leaderboards in a workplace and found that they significantly improved staff motivation through social comparison. Supporting this, Landers and colleagues [24] reported that leaderboards enhanced the completion of a short-term task. Meanwhile, a study on gamified image tagging platforms showed that leaderboards did not positively influence intrinsic motivation [69]. From the literature, we infer that the leaderboard will positively affect extrinsic motivation to continue using gamified fitness apps. Accordingly, it is proposed that:

**H11.** 
*Leaderboards have a positive effect on extrinsic motivation to continue using gamified mHealth apps.*


### 2.14. The Effect of Social Comparison (SC)

Social psychologists [32] have established how motivation affects human behavior through social comparison. The theory describes how users’ motivation can be increased through social comparison and self-enhancement. People seek informative feedback about their abilities by comparisons with others [32]. Consequently, this often leads to competition among peers, ultimately motivating them. In the current IS literature, social comparison was extensively proven to be a strong determinant of user motivation to engage in physical activity [24,40]. In addition, social comparison is leveraged through multiple game elements such as leaderboards, badges, and feedback [25]. This suggests that when individuals enjoy competing against their peers, they tend to enhance their physical activity. Eventually, users will likely sustain engagement with mHealth to improve their performances. We expect social comparison to motivate users’ intention to continue using mHealth extrinsically; hence, the following hypothesis is proposed:

**H12.** 
*Social Comparison has a positive effect on extrinsic motivation to continue using gamified mHealth apps.*


### 2.15. Intrinsic Motivation (IM)

Intrinsic motivation pertains to performing a task because it is personally rewarding [39,70]. A classic example is playing sports for enjoyment instead of doing it for a reward. Intrinsically motivated individuals engage in activities because they are fun, challenging, engaging, or captivating [44]. This type of motivation can drive behavior in sports or exercise. IM is improved when feelings of competence and self-determination are enhanced [41]. Conversely, IM diminishes when the autonomy level is low. Previous empirical has continuously identified motivation as a significant predictor of user adherence to physical activity [40,71]. For instance, when individuals exercise for internal reasons, they are more likely to be fulfilled and sustain engagement for extended periods. Thus the satisfaction of innate needs positively correlates with IM. Gamified fitness apps introduce game-like features, including challenges, avatars, or groups to encourage and sustain exercising [72]. Game-like features must produce a long-lasting commitment to exercise. Based on the existing literature, this study assumes that intrinsic motivation can significantly impact the continued use of mHealth, leading to the following hypothesis:

**H13.** 
*Intrinsic motivation positively affects the continued use of gamified mHealth apps.*


### 2.16. Extrinsic Motivation (EM)

Extrinsic motivation denotes performing an activity to achieve an outcome or reward [42]. EM is driven by the prospect of gaining benefits or avoiding negative consequences. For example, people engage in sports for a medal or reward. A reward is an extrinsic motivator that can motivate humans, but the effect is short-term and unsustainable [73]. While IM focuses on internal benefits, EM centers on external rewards. In gamified mHealth, EM is essential in incentivizing individuals to start exercising [41]. This is achieved through game-like elements such as leaderboards, levels, badges, points, or rewards. However, there is a need for caution as prior studies indicate that EM can have detrimental effects on IM and undermine gamification designed to enhance IM [37]. That is, extrinsically motivated individuals may experience less pleasure from performing activities and become uninspired or demotivated to continue engaging. EM for physical activity participation can also contribute to lower levels of commitment to regular exercise [41]. Concerning this, it is critical to understand the effect of EM on sustained engagement in gamified mHealth. Thus, the following is put forward:

**H14.** 
*Extrinsic motivation negatively affects the continued use of gamified mHealth apps.*


### 2.17. Facilitating Conditions (FC)

The IS body of literature identifies facilitating conditions as critical technical support for performing a behavior [74]. However, our study adopts the definition in the mHealth literature. In this sense, researchers define facilitating conditions (FC) as an environmental factor that could significantly influence physical activity participation [25,75]. This represents the resources and support required to perform a task, such as time, funds, access to facilities, and other specific resources [25]. Some studies reported that the environment determines healthy behaviors [25]. The environment can influence individuals by promoting or preventing physical activity, such as walking, jogging, or running. In the context of this study, FC fosters conditions related to the social (social media platform) and physical environment (weather and facilities) for exercising [75]. To this end, the study assumes that facilitating conditions affect the continued use of gamified mHealth. Thus, the following hypothesis is proposed:

**H15.** 
*Facilitating conditions are positively related to the continued use of gamified mHealth apps.*


## 3. Instrument Development

The research model theoretically comprises fifteen constructs primarily based on SDT and ScT. Five constructs were drawn from gamification, while the extant literature theoretically guides other variables. All latent variables were adopted from prior studies and adapted to fit the gamified mHealth context. Before developing the measurement instruments for the proposed model, we reviewed the existing literature to establish a comprehensive list of measures from existing validated scales. The study adopted perceived autonomy, competence, and relatedness measures from SDT [31]. The social comparison construct was adopted from ScT [25,32], whereas levels, progress bars, feedback, challenges, and leaderboards were from the gamification literature and a preliminary study [58,76]. Other measures from prior literature include perceived benefits [54], perceived hedonic gratification [77], fitness app quality [78], and facilitating conditions [25]. Finally, intrinsic and extrinsic motivations were drawn from the motivational context [37], specifically the self-determination theory domain. The justification for each construct is discussed in Section 2. Table 1 illustrates the variables of the proposed model, including the latent variables associated with them and their sources.

**Table 1 healthcare-11-01051-t001:** Sources of Items.

Construct	Definition	Source
Facilitating Conditions (FC)	The conditions that facilitate the use of fitness apps for exercise.	[25]
Social Comparison (SC)	The degree to which users self-evaluate their exercise performance with others.	[25]
Perceived Autonomy (PA)	The need for a user to feel that they can choose their actions without pressure.	[39]
Perceived Competence (PC)	The need to successfully achieve desired results and avoid unwanted outcomes.	[39]
Perceived Relatedness (PR)	The need to feel connected with others and be valued by them.	[39]
Intrinsic Motivation (IM)	The degree to which an individual is driven to use a mHealth for internal rewards.	[39]
Perceived Benefits (PB)	The degree to which the user perceives the mHealth will enhance their performance.	[54]
Perceived Hedonic Gratification (PHG)	The degree to which the process of using mHealth is perceived to be pleasant.	[77]
Fitness App Quality (FAQ)	The degree to which the fitness app is easy to use, reliable, functional, and efficient.	[78]
Feedback (FB)	Personalized real-time updates about users’ performance.	[79]
Challenge (CH)	A task that requires considerable effort to solve.	[79]
Progress Bars (PGB)	A visual representation of users’ progress toward a goal.	[79]
Leaderboards (LB)	Visual display of users’ rankings based on their achievements.	[79]
Level (LV)	Indication of user’s progress over time.	[79]
Extrinsic Motivation (EM)	The degree to which an individual is driven to use a mHealth for external rewards.	[80]
Continued Use (CU)	The degree to which a user intends to continue using mHealth.	[81]

### Construct Conceptualization

Conceptualization presents a clear conceptual definition of a construct with a single interpretation [82]. The constructs should have unambiguous terms and be consistent with prior studies [82]. Importantly, it should be distinguished from related constructs. Table 1 presents definitions for the constructs in this study. The definitions were adapted to fit the study purpose.

## 4. Methodology

In this study, we assessed the proposed research model for reliability with data collected via a survey conducted among gamified mobile health users in Malaysia. Instrument validation is essential in empirical IS research [83]. According to [84], instruments adopted from the literature should be validated. This is because the adopted items may have needed to be effectively evaluated. In addition, the modified items could lose their original validity. The pilot study evaluated the survey instrument’s internal consistency reliability (Cronbach’s alpha and item-to-total correlations). The methodology includes a content validity and reliability test.

### 4.1. Study Design, Sampling Population, and Sample Size

Simple random sampling was used in the study. According to [85], this falls within the range of acceptable sample size, using 90% power and 5% significance. An online-based survey was employed as it is convenient, more accessible, and cost-effective. The users of gamified mHealth were invited to complete the questionnaire voluntarily without any reward.

### 4.2. Inclusion Criteria

Eligibility criteria involved individuals currently using mHealth apps to engage in physical activity, aged 18 years and above, and residents in Malaysia.

### 4.3. Survey Instrument: Questionnaire

Questionnaire design is vital to the data collection process. We developed the items to measure the constructs and operationalize the proposed research model. The scale items in the survey were mainly from prior literature and adapted to fit the gamified mHealth context. The self-administered questionnaire comprised four sections. Section 1 covers demographic questions, Section 2 covers user perception (six items), Section 3 covers game elements (five items), and Section 4 covers user motivational factors (five items). Moreover, a 7-point Likert scale of 1 (“strongly disagree”) to 7 (“strongly agree”) was employed to measure the latent variables. Further, the questionnaire was translated into Bahasa Malaysia (the local language) for respondents where, English was not their preferred language.

### 4.4. Data Collection

We used a cross-sectional survey research method to collect quantitative data. Eligible respondents were recruited online and requested to complete the online-based survey. Data collection took place in November 2021 through Google forms. Screening questions were applied to confirm that only current fitness app users participated in the survey. Further, participants were oriented about the research purpose and definition of study variables before the screening questions were displayed. After data screening, 48 valid data were collected from fitness app users in 12 states across Malaysia. Most respondents were from Selangor, followed by Kuala Lumpur and Johor. No issues that prevented any of the participants from completing the survey were reported.

### 4.5. Data Analysis

The SPSS Version 22 statistical software and SmartPLS4 were used to organize and analyze the data, and descriptive statistics were employed to establish sociodemographic characteristics.

### 4.6. Study Ethics

The Medical Research and Ethics Committee of the Ministry of Health, Malaysia approved this study (NMRR ID-22-00339-ZBN) and all respondents gave informed consent. The research procedures followed all appropriate national and international standards.

## 5. Results

### 5.1. Analysis

In this study, 48 individuals completed the online-based survey with a response rate of (80%). Data collection for the pilot study concluded in December 2021. Table 2 summarizes the respondent’s demographic data. The mean age was 26.9 years, with 58% females (*n* = 28). Most respondents were from Selangor (33%), Kuala Lumpur (15%), and Johor (13%). Similar to previous studies [86], the vast majority of respondents (50%, *n* = 24) have an undergraduate degree. Most participants reported being employed (52%), while 48% were students. Notably, the participants utilized 15 different mHealth apps. The five most prominent, as depicted in Table 2, include MyFitnessPal (15%), Strava (12.5%), Samsung Health (10.5%), RunKeeper (10.5%), and HealthifyMe (10.5%). 

### 5.2. Face Validity

We consulted four experts in mHealth and gamification to assess the instrument’s validity. They carefully reviewed the survey instrument for clarity and understandability. The psychologist and researchers were familiar with the concept under investigation and instrumentation. The latent and observed variables were assessed based on complications, ambiguity, relativity, proper terms and grammar, and understandability. Several minor mistakes detected, including spelling errors and duplication, were amended accordingly to make the questionnaire easier to understand. After some grammatical alterations, the survey was finalized.

### 5.3. Content Validity

As a next step, the content validity of the measurement instruments was tested using a formal content validity test to ensure that measurement items reflect all of the critical aspects of their constructs [87]. Content validity involves qualitative and quantitative phases; however, this study employed the quantitative approach for objectivity. Three gamification experts familiar with psychometric methods were involved in this process. A content validity question was developed based on the four-point rating scale of [88] and question appraisal system (QAS) [89]. The QAS effectively assesses cognitive issues that might arise when completing the questionnaire. The instrument item relevance section uses a scale from 1 (“not relevant”) to 4 (“highly relevant”) to rate each statement’s relevance. In the second section (QAS), experts were requested to assess the statements based on eight criteria: instruction, recall, sensitivity/bias, clarity, response option, format, content coverage, and length [89]. The initial instrument contained 42 items, excluding the gamification constructs. 

The responses were calculated using the Average Congruence Percentage (ACP) based on the methods described by [87]. Overall, 83% of the content validity study items met the criterion of 0.9 [87]. Based on the content validity results, items from facilitating conditions and continued use were replaced. The perceived competitive climate and social comparison construct were also merged, while two items were dropped from social compassion. Although one item of perceived relatedness was removed, the other item that did not meet the criteria was retained as it was considered significant based on the literature. Finally, an academic language expert translated the survey item statements from English to Bahasa Malaysia using forward and backward translation for consistency.

### 5.4. Reliability

The reliability of the variables was tested using Cronbach’s Alpha (α) and Composite Reliability (CR). Cronbach’s alpha is the most appropriate measure of reliability when using the Likert scale, with values ranging from 0 to 1; a higher range level shows greater reliability [90]. For reliability, it is suggested that values above 0.9 implies excellent, 0.70 to 0.90 signifies high, 0.50 to 0.70 shows moderate, and 0.50 indicates low reliability [91]. The internal consistency reliability of each construct’s items was tested using SmartPLS 4. The results for reliability and validity for the remaining items are presented in Table 3. Thus, all the constructs in the model achieved alpha and CR values above the criterion of 0.7, suggesting a suitable construct reliability [91]. To increase reliability, the “if item deleted” function was applied to identify the item with the lowest correlation with other items in the same group [92]. Based on the reliability analysis results, PR3 was deleted to improve the alpha value of perceived relatedness from 9.18 to 9.24. In this study, we started with 59 items; based on the validation process, ten items were removed, and three items were replaced, resulting in 52 items (88%) believed to be valid.

## 6. Discussion

The present study aimed to develop and validate a quantitative measurement instrument for the proposed research model to identify factors affecting the sustained use of gamified mHealth. Because no validated instrument fits this investigation’s purpose, this instrument was developed. In addition, the instrument was validated through a pilot study. Most notably, instrument validation is significant in determining its reliability and validity before conducting the main study [93]. It also increases confidence in the collected data. According to the literature, a pilot study is commonly used to validate a measurement instrument in a study [83]. Further, prior studies [84] stress the importance of face and content validity before a pilot study.

The face validity findings suggested amending some items’ choice of words and improving clarity. Accordingly, the experts’ recommendations improved the instrument’s relevance and appropriateness. The content validity test resulted in the removal of 9 items. Thus, it was concluded that the remaining items in the measurement instruments fit the context of mHealth and accurately reflected the integrated model’s constructs. The results also suggest that the question appraisal system (QAS) is useful for the content validity of the measurement instrument in a quantitative study. Other studies employed the content validity index (CVI) to assess content validity [87]. However, the QAS examines the sensitivity and social acceptability of the questions [89]. This is important in health- and fitness-related studies where the wording is important. It is also beneficial for questions involving cross-cultural populations. The QAS results show that 83% of all items in the instruments were acceptable. Therefore, those items were considered suitable for inclusion in the main study.

All the model’s constructs achieved composite reliability values above 0.7. In line with this, past studies recommend a reliability of 0.6 or above [83]. This finding is consistent with previous pilot studies [94]. Hence, the result implies a suitable construct reliability [91]. Notably, reliability should be combined with validity to ensure the instrument is accurate and consistent [95]. Hence, this study applied the two techniques (reliability and validity). Overall, the reliability values reflect excellent instrument validity and reliability. Out of the 59 items (see Appendix A), ten were removed, three were replaced, and 52 were retained. In addition, one item (PR3) was removed based on the pilot study results. This study also confirmed that most items are reliable. Hence, we imply that the content and face validity improved the validity and reliability of the instrument to a satisfactory level. 

In sum, the results are consistent with [35], which applied a similar procedure to confirm the feasibility of measurement instruments in the mHealth domain. Most importantly, accuracy and consistency of a survey questionnaire form a significant aspect of the research methodology. This case has shown that this study’s measurement instrument is suitable for assessing the sustained use of gamified mHealth for promoting physical activity. Therefore, based on the study findings, the main study is feasible.

## 7. Conclusions

Given the considerable number of studies on gamification, several studies are grounded in a single theory and focus on user adoption [96]. The existing body of literature mainly focuses on gamification in the education domain [97]. We identified factors related to the user’s demographics, personal information, and use of mHealth that can potentially affect the sustained engagement of gamified mHealth. Nonetheless, those factors may not significantly impact the continued intention, as the study sample is current fitness app users. Research on theoretical integration is attracting attention in the IS literature [71,98]. However, integrated models are employed in a few gamification studies [99]. The nature of the factors in the proposed model highlights its suitability for application within the gamified mHealth context. Therefore, the integrated model aims to offer valuable insights into understanding users’ continued usage from different perspectives and expands the theoretical knowledge base of the sustained use of mHealth. 

This study’s findings offer some contributions. First, we proposed an integrated theoretical model for an in-depth understanding of the effects of game elements in sustained engagement with mHealth. The integrated model retains the essential components of each model and supplements each other. Despite its similarity to existing models [39,72,100], the proposed model addresses factors related to two motivational styles (extrinsic and intrinsic) in gamified mHealth. The integrated model is a more advanced framework than the SDT alone, with external variables complementing it. Essentially, the integrated model offers a deeper understanding of users’ long-term engagement with mHealth, incorporating game-design elements. Second, we developed a survey instrument to measure users’ adherence to gamified mHealth. This process involved assessing existing instruments from the extant literature and modifying items to fit the gamified mHealth context. The systematic approach to the survey instrument development offers high confidence in its construct validity. Third, the proposed model contains aspects based on game features reported in the literature and a preliminary study. Accordingly, this model presents a compelling argument for theory integration and instrument testing while identifying significant implications for mHealth developers. From a theoretical viewpoint, this study verifies that the developed instrument is valid for investigating the continued use of gamified mHealth. Finally, this study instrument’s high degree of reliability and validity provides a foundation for further research.

### Limitations and Future Directions

There are some limitations to this study. This study used a sample size of 46. While this is considered satisfactory, similar studies [94] used more significant sample sizes. However, other studies used smaller sample sizes [101]. This study should have requested participants’ feedback regarding the readability and clarity of the questions/statements. In line with existing recommendations [102], subjects need to provide feedback to identify ambiguities and challenging questions. Therefore, it was not feasible for the researchers to receive valuable feedback from respondents. In addition, this study should have observed the time it takes to complete the survey. The questionnaire completion time assists the researcher in deciding whether it is reasonable. Questionnaires that require a long time to complete can negatively impact the response rate. It can also increase survey abandonment rates. Thus, future pilot studies should consider taking note of the questionnaire completion time. Moreover, some authors [102] suggest conducting multiple pilot studies (testing and re-testing) to increase reliability. Equally important, when sending a developed questionnaire with the measurement scale to expert reviewers, researchers should provide relevant information to the evaluators to ensure appropriate feedback.

## Figures and Tables

**Figure 1 healthcare-11-01051-f001:**
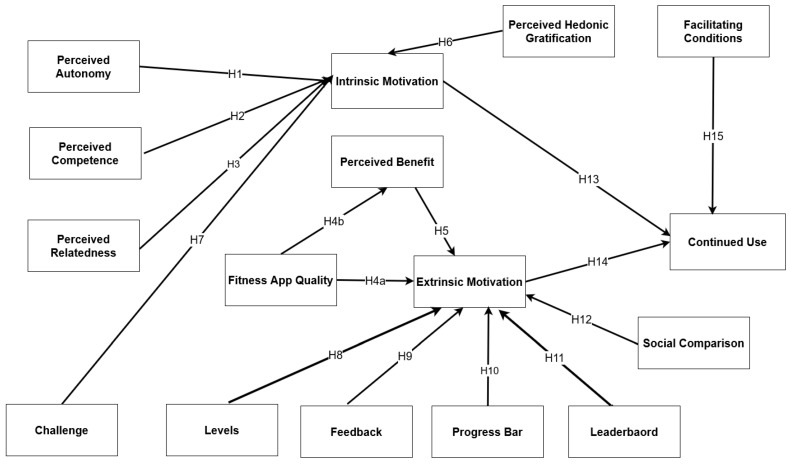
Proposed Model.

**Table 2 healthcare-11-01051-t002:** Demographic information of the respondents.

	Frequency	Percentage
Gender		
Male	20	42
Female	28	58
Age (Mean = 26.9)		
18–25	28	58
26–35	12	25
36–45	8	17
Educational Level		
High School	5	10.5
Diploma	5	10.5
Bachelors’ Degree	24	50
Masters’ Degree	12	25
PhD Degree	2	4
Monthly Household Income		
Less than 2000	20	42
2999–4999	9	18.5
5000–7999	8	17
8000–9999	2	4
Above 10,000	9	18.5
Frequently Used Apps		
MyFitnessPal	7	25
Strava	6	21.4
Samsung Health	5	17.9
RunKeeper	5	17.9
HealthifyMe	5	17.9

**Table 3 healthcare-11-01051-t003:** Reliability analysis for each construct.

Construct	Item	No. of Items	Cronbach’s Alpha	Composite Reliability
Perceived Autonomy (PA)	PA1, PA2, PA3, PA4	4	0.929	0.949
Perceived Competence (PC)	PC1, PC2, PC3, PC4	4	0.931	0.951
Perceived Relatedness (PR)	PR1, PR2, PR4 (PR3 deleted)	3	0.919	0.943
Fitness App Quality (FAQ)	FAQ1, FAQ2, FAQ3	3	0.920	0.950
Perceived Benefits (PB)	PB1, PB2, PB3	3	0.950	0.968
Perceived Hedonic Gratification (PHG)	PHG1, PHG2, PHG3	4	0.938	0.955
Leaderboard (LB)	LB1, LB2, LB3, LB4	4	0.884	0.916
Levels (LV)	LV1, LV2, LV3, LV4	4	0. 868	0.908
Feedback (FB)	FB1, FB2, FB3, FB4	4	0.849	0.906
Progress Bar (PGB)	PGB1, PGB2, PGB3, PGB4	4	0.950	0.968
Challenge (CH)	CH1, CH2, CH3, CH4	4	0.922	0.945
Social Comparison (SC)	SC1, SC2, SC3, SC4	4	0.939	0.956
Facilitating Condition (FC)	FC1, FC2, FC3	3	0.924	0.952
Intrinsic Motivation (IM)	IM1, IM2, IM3	3	0.924	0.952
Extrinsic Motivation (EM)	EM1, EM2, EM3, EM4	4	0.893	0.925
Continued Use (CU)	CU1, CU2, CU3	4	0.949	0.963

## Data Availability

Not applicable.

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
