# Peer review of "An Integrated Model for Evaluating the Sustainability of Gamified Mobile Health Apps: An Instrument Development and Validation"

_healthcare, 2023, doi:10.3390/healthcare11071051_

Round 1

Reviewer 1 Report

The authors presented an interesting model to develop more sustainable mHealth apps. The paper is interesting and well written but several points should be addressed before the paper can be accepted. It's not clear what type of article it is? The methodology should be more described and the paper should be restructured according to current standard.

- This study, and app, is about obesity management. It should be stated in the title as the situation would be totally different for other apps targeting different population and objectives (e.g., dementia, stroke patients, children with CP, etc.).

- mHealth is (much) broader than games and gamification (e.g., calendar, health literacy, social aspects), please adapt the abstract and manuscript accordingly and provide a description and definition of mHealth and the different field of applications before focusing on the games.

- Over the research model (section 2): the authors did a long list of the different factors, how did you select those? How did you build the Figure 1?

- Method: 2.2. why include people using mHealth aged above 18 and not focusng on obesity management? Or at least app to engage in physical activity? What is the mean BMI and physical activity levels of the participants? 

- Figure 2 does not bring anything, info is presented in Table 2.

- Table 3: Where can we find the definition of the different items presented in the different categories? 

- The discussion is barely not a discussion, the authors did not cite any reference nor discuss the content of other studies in the 2 first paragraph! This should be much more extended. Limitations should also be presented as well as the implications for the care. 

Author Response

Thank you for allowing us to submit a revised draft of our manuscript titled "An Integrated Model for Evaluating the Sustainability of Gamified Mobile Health Apps: An Instrument Development and Validation."

Comment 1: The authors presented an interesting model to develop more sustainable mHealth apps. The paper is interesting and well written but several points should be addressed before the paper can be accepted. It's not clear what type of article it is? The methodology should be more described and the paper should be restructured according to current standard.

Response: Thank you for pointing this out. We agree with these recommendations. Based on MDPI standards, the manuscript has been categorized as an Article. As suggested, we have revised the methodology in section 4 (pages 11 and 12).

Comment 2: This study, and app, is about obesity management. It should be stated in the title as the situation would be totally different for other apps targeting different population and objectives (e.g., dementia, stroke patients, children with CP, etc.).

Response: Thank you for this suggestion. The focus of the apps is to encourage physical activity or regular exercise. Obesity is one of the risk factors for physical inactivity. Thus, obesity is not the main focus of the app. However, it is expected that physically active individuals are less likely to be overweight or obese. For clarity, we have revised the manuscript to highlight that the study focuses on the continued use of fitness apps for promoting physical activity.

Comment 3:  mHealth is (much) broader than games and gamification (e.g., calendar, health literacy, social aspects), please adapt the abstract and manuscript accordingly and provide a description and definition of mHealth and the different field of applications before focusing on the games.

Response: Agree. We have, accordingly, revised the Introduction (Page 1, Section 1) to emphasize this point. A broader definition has been provided on Page 2 (lines 49-68). The concept of gamification involves incorporating game elements in another context. Existing health apps can also be gamified (incorporate game elements).

Comment 4:  Over the research model (section 2): the authors did a long list of the different factors, how did you select those? How did you build the Figure 1?

Response: Thank you, we have already reflected this in Section 2 (Pages 3 to 5). Also, in section 2, we have justified the selection of each construct in the model under subsections.

Comment 5:  Method: 2.2. why include people using mHealth aged above 18 and not focusing on obesity management? Or at least app to engage in physical activity? What is the mean BMI and physical activity levels of the participants?

Response: Thank you for pointing this out. The focus is on using the app to sustain physical activity. The assumption is that the continued use of the app will encourage people to exercise regularly. Therefore, we have highlighted this within the manuscript.

Comment 5:  Figure 2 does not bring anything, info is presented in Table 2.

Response: Thank you for pointing this out. We removed figure 2 because it repeats the information in Table 2.

Comment 6:  Table 3: Where can we find the definition of the different items presented in the different categories?

Response: Thank you for this suggestion. The definition of the items is included in Table 1 (page 11).

Comment 7:  The discussion is barely not a discussion, the authors did not cite any reference nor discuss the content of other studies in the 2 first paragraph! This should be much more extended. Limitations should also be presented as well as the implications for the care.

Response: We agree with this suggestion. Therefore, we have rewritten Section 6 (page 15, line 601) to discuss the results. Limitations and future directions are included in Section 7.1.

Reviewer 2 Report

I congratulate the Authors on the idea to conduct a literature review and construct a pilot study.

Authors set specific goals and achieved them. Since this pilot can be described as validation of the author's questionnaire, it would be useful to indicate what percentage of questions raised doubts and whether they have changed. In addition, it may be worthwhile to indicate recommendations for further use of the developed questionnaire in the conclusions.

Author Response

Thank you for allowing us to submit a revised draft of our manuscript titled "An Integrated Model for Evaluating the Sustainability of Gamified Mobile Health Apps: An Instrument Development and Validation."

Response: Thank you for pointing this out. We agree with this comment. Unfortunately, the study did not request feedback from the subjects. This would have been valuable to the researchers. However, we assumed that the feedback from experts was sufficient. Based on the content validity, 88% of the questions were retained (reflected on Page 14, line 597).

Response: Regarding recommendations for future study, we appreciate this critical point and include Section 7.1 to address limitations and recommendations (future directions).

Reviewer 3 Report

Dear authors,

In my opinion, section number 6, Discussion, is too short and not sufficient to explain the data you have collected.

In my opinion, the style of Section 6 is more similar to that of a conclusion than to that of a discussion.

You should better arrange the text of your paper and better help the reader to understand the results you have obtained from your study by expanding on what you have written and moving some content from section 5 to section 6.

Wouldn't it be useful to put a small example of what the questionnaire looks like?

Author Response

Thank you for allowing us to submit a revised draft of our manuscript titled "An Integrated Model for Evaluating the Sustainability of Gamified Mobile Health Apps: An Instrument Development and Validation."

Comments 1, 2, and 3: In my opinion, section number 6, Discussion, is too short and not sufficient to explain the data you have collected.

In my opinion, the style of Section 6 is more similar to that of a conclusion than to that of a discussion.

You should better arrange the text of your paper and better help the reader to understand the results you have obtained from your study by expanding on what you have written and moving some content from section 5 to section 6.

Response: We agree with this point. Accordingly, we have modified sections 5 and 6 to emphasize this point and for clarity.

Section 5 focuses on results, while Section 6 on the discussion.

Comment 4: Wouldn't it be useful to put a small example of what the questionnaire looks like?

Response: Thank you for this suggestion. We have incorporated this suggestion into the manuscript in the Appendix (Appendix A.1.)

Round 2

Reviewer 3 Report

The authors have made the suggested improvements